# Mixture of Heterogeneous Grouped Experts for Language Modeling

## Abstract

Mixture-of-Experts (MoE) offers superior performance over dense models. However, current MoEs impose a critical limitation by enforcing uniform expert sizes, restricting the model's ability to dynamically match computational resources with token-specific requirements. Despite several attempts on heterogeneous experts have been made, they struggle either with limited performance and inefficient parameter utilization or unbalanced GPU utilization, there is still a lack of general heterogeneous MoE architecture. To this end, we present Mixture of Heterogeneous Grouped Experts (MoHGE), an innovative MoE architecture that introduces a two-level routing mechanism and enables more nuanced and efficient expert selection tailored to each input token's characteristics. We also propose a Group-Wise Auxiliary Loss to enhance efficient parameter utilization without compromising model performance. To address the resulted workload imbalance challenges, we develop: (1) an All-size Group-decoupling Allocation strategy and (2) Intra-Group Experts Auxiliary Loss, collectively ensuring balanced GPU utilization. Extensive evaluations on multiple benchmarks demonstrate that MoHGE achieves comparable performance to state-of-the-art MoE architectures while reducing total parameter count by approximately 20% and maintaining balanced GPU utilization. Our work establishes a new paradigm for resource-aware MoE design, better aligning computational allocation with actual inference demands.

## 1 Introduction

Transformer-based large language models (LLMs) (Achiam et al., 2023; Touvron et al., 2023; Bai et al., 2023; Liu et al., 2024a) have achieved remarkable success across a wide range of natural language processing (NLP) tasks. According to scaling laws (Kaplan et al., 2020), larger models consistently deliver better performance, and recent studies (Wei et al., 2022) have shown that scaling can also give rise to emergent abilities. However, the computational cost of training and deploying such large models grows exponentially (Thompson et al., 2020), creating a critical bottleneck for both research and real-world applications.

Mixture-of-Experts (MoE) architectures, originally proposed in Jacobs et al. (1991) and Jordan & Jacobs (1994), offer an effective solution by enabling sparse activation: Only a small subset of the model parameters are engaged in per inference step, allowing the model to scale efficiently without proportionally increasing computational overhead.

Despite this advantage, most existing MoE models consist of experts with identical sizes and structures. This homogeneity poses a limitation when generating tokens of varying difficulty: some tokens are easy to predict, while others require more sophisticated reasoning. To address this, recent approaches such as MoDSE (Sun et al., 2024) and HMoE (Wang et al., 2024) have explored the use of experts with different sizes.

However, MoDSE employs a routing strategy that promotes uniform routing probabilities among experts, which fail to route input tokens to the most suitable experts, leading to inefficient parameter utilization. Since experts have different parameter sizes, this setting limits the combination of experts, making it impossible to select multiple smallest or largest experts, missing opportunities for better efficiency or performance. HMoE mentions the idea of hybrid heterogeneous–homogeneous experts as a promising direction, but does not explicitly explore this design. Moreover, it suffers

from significant GPU utilization imbalance due to uneven parameter sizes, ultimately degrading training efficiency and limiting its scalability.

In this paper, we first divide experts into multiple groups, where experts within each group share identical parameter sizes, while the expert sizes vary across groups. We then introduce a two-level routing strategy to deliver more diverse and nuanced expert combinations. We further propose a Group-Wise Auxiliary Loss to enable the selection of expert groups with appropriate parameter sizes, based on the task difficulty. This ensures more efficient parameter utilization by dynamically matching computational resources to token-specific requirements. To address GPU load imbalance, we propose an All-size Group-decoupling Allocation strategy, which places an equal number of experts from each group onto the each GPU. This strategy guarantees that each GPU has the same memory consumption. Further, we propose an Intra-Group Experts Auxiliary Loss to maintain balanced routing probabilities within each expert group, ensuring uniform GPU utilization. We refer to this novel architecture as the Mixture of Heterogeneous Grouped Experts (MoHGE). Our contributions are summarized as follows:

- **Novel Architecture**: We propose a novel MoE architecture, MoHGE, that achieves precise capacity match based on task difficulty and efficient GPU utilization by incorporating the two-level routing strategy and the Group-Wise Auxiliary Loss.

- **Load Balance**: To ensure balanced GPU utilization, we propose the All-size Group-decoupling Allocation strategy and the Intra-Group Experts Auxiliary Loss. Together, these techniques maintain intra-group utilization equilibrium and achieve uniform GPU workloads, ensuring the model's scalability.

- **Empirical Validation**: Experimental results demonstrate the framework's effectiveness: MoHGE achieves an accuracy comparable to that of conventional MoE while reducing total parameters. More noteworthy, detailed routing analysis confirms successful balance of GPU utilization and validates our loss functions' ability to regulate expert activation patterns.

## 2 BACKGROUND: MIXTURE OF EXPERTS

An MoE layer typically includes the gating model $G_1(\cdot) \cdots G_N(\cdot)$, the expert networks $E_1(\cdot) \cdots E_N(\cdot)$, and the routing mechanism, where $N$ denotes the number of experts. The gating model serves as the mathematical implementation of a router, determining how input data is allocated to experts. Specifically, the gating model with learnable weights $W \in \mathbb{R}^{h_{input} \times h}$ selects the top $k$ experts and combines the outputs of these top $k$ experts to produce the output $y \in \mathbb{R}^h$, where $h_{input}$ is the dimension of input $x$ and $h$ is the dimension of the hidden layer. The output of an MoE layer can be expressed as,

$$y = \sum_{i=1}^{N} G_i(x) E_i(x) \tag{1}$$

$$G_i(x) = Softmax(topK(H(x))) \tag{2}$$

$$H(X)_i = (x \cdot W)_i \tag{3}$$

$$TopK(v, k)_i = \begin{cases} v_i, & v_i \in topk(v) \\ -\infty, & \text{otherwise} \end{cases} \tag{4}$$

## 3 MIXTURE OF HETEROGENEOUS GROUPED EXPERTS

### 3.1 GROUP-WISE VARIED SIZE EXPERTS

Traditional MoE architectures typically employ a gating network that routes inputs to a uniform set of experts, all of which have the same model size. However, as shown by Sun et al. (2024), the cognitive challenge of predicting the next token varies significantly across different linguistic contexts—mirroring the dynamic processing demands seen in human cognition.

Building on this observation, we introduce a novel heterogeneous expert architecture that organizes experts into multi-granularity groups. Formally, we structure the expert set $\{E_1, E_2, E_3, \cdots, E_{N_e}\}$

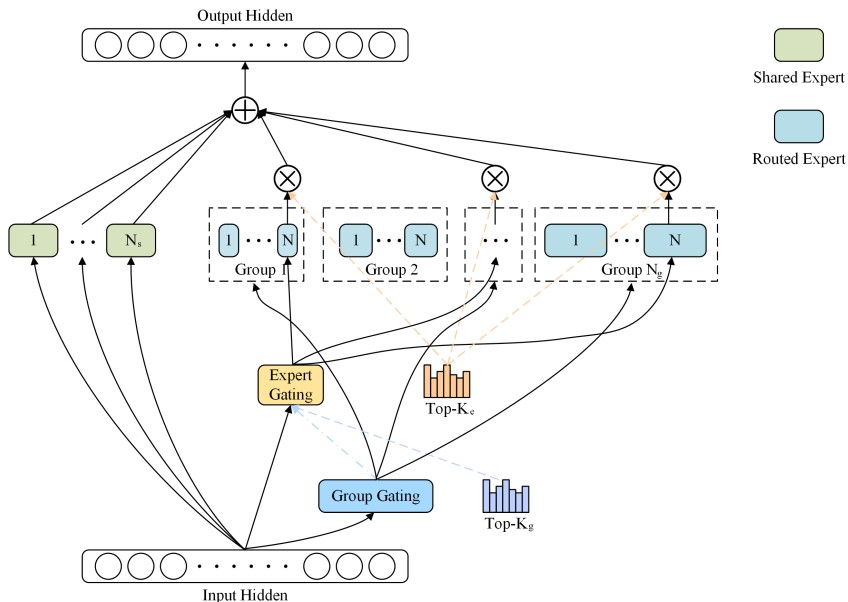

Figure 1: An illustration of our Mixture of Heterogeneous Grouped Experts Layer.

into distinct groups $\{G_1, G_2, G_3, \cdots, G_{N_g}\}$, where each group contains $N = N_e/N_g$ experts ($N_e$ and $N_g$ denote the total number of experts and groups, respectively). For the convenience of expression, we transform experts from $E_j$ into $E_{g,i}$, where $g$ represents the group to which the expert belongs and $i$ represents the index of the expert in the group. Experts within each group share identical parameter sizes, while parameter scales vary across groups according to a predefined progression. Specifically, the hidden dimension of experts in group $G_i$ is given by:

$$W_i = 2 * W_{\text{base}} - W_{N_g - i} \tag{5}$$

where $W_{\text{base}}$ represents the base hidden dimension and the $W_i$ increases as $i$ increases. This hierarchical organization enables dynamic computation allocation: compact experts efficiently process simpler linguistic patterns, while progressively larger experts with greater capacity handle more complex contextual relationships.

## 3.2 TWO-LEVEL ROUTING MECHANISM

To efficiently manage the hierarchical structure of experts, our two-level routing mechanism operates in two stages. The **group gating model** first selects expert groups based on their relevance to the input, and the **expert gating model** then chooses specific experts within these groups. This staged design ensures that computation is focused on the most relevant experts, reducing overhead by restricting selection to the top-$K_g$ groups.

### 3.2.1 GROUP GATING MODEL

The group gating model computes scores $GS$ for all $N_g$ expert groups. For the $t$-th token input $\mathbf{x}_t$, the score for the $g$-th group is,

$$GS_{g,t} = Sigmoid(\mathbf{x}_t^T \mathbf{e}_g) \tag{6}$$

where $\mathbf{e}_g$ is the centroid embedding of the $g$-th expert group. The model then selects the $K_g$ groups with the highest scores, restricting the expert gating model to only route tokens to experts within these groups.

### 3.2.2 EXPERT GATING MODEL

The expert gating model operates in three phases: **Intra-Group Expert Scores Calculation**, **Experts for Global Selection** and **Global Normalization**.

**1. Intra-Group Expert Scores Calculation.** For each selected group, the model computes unnormalized scores for its experts using a group-wise Softmax:

$$ES'_{g,i,t} = \begin{cases} \mathrm{Softmax}(\mathbf{x_t}^T \mathbf{e}_{g,i}), & \text{if } GS_{g,t} \in \mathrm{top}K_g(GS_t) \\ 0, & \text{otherwise} \end{cases} \tag{7}$$

where $e_{g,i}$ is the embedding of the $i$-th expert in group $g$.

**2. Experts for Global Selection.** The intra-group expert scores are scaled by the group scores to reflect group importance:

$$ES''_{g,i,t} = (ES' \cdot GS)_{g,i,t} \tag{8}$$

Next, the model selects the top-$K_e$ experts globally. Scores for all other experts are set to zero:

$$ES'''_{g,i,t} = \begin{cases} ES''_{g,i,t}, & \text{if } ES''_{g,i,t} \in \mathrm{top}K_e(ES''_{g,i,t}) \\ 0, & \text{otherwise} \end{cases} \tag{9}$$

**3. Global Normalization.** Finally, the selected expert scores are normalized to sum to one:

$$ES_{g,i,t} = \frac{ES'''_{g,i,t}}{\sum_j^{N_g} \sum_k^N ES'''_{j,k,t}} \tag{10}$$

This three-step gating strategy enables fine-grained, efficient expert selection by prioritizing both group relevance and individual expert utility.

### 3.3 OUTPUT OF MoHGE

The output of MoHGE layer is similar to the MoE layer, the outputs of all selected experts are multiplied by their corresponding scores and then added together to obtain the final output:

$$y = \sum_{g=1}^{N_g} \sum_{i=1}^{N} ES_{g,i,t} \cdot E_{g,i}(x_t) \tag{11}$$

### 3.4 EFFICIENT PARAMETER UTILIZATION

Without regularization, experts with larger parameter sizes tend to dominate the routing decisions due to their stronger representational capacity. This dominance can result in inefficient expert usage, as smaller expert groups with fewer parameters may not be fully utilized. To address this issue and improve parameter utilization, we introduce a slight penalty for expert groups with larger parameter sizes. Specifically, we propose **Group-Wise Auxiliary Loss** $L_G$, which slightly penalizes expert groups with larger parameter sizes.

This loss encourages the gating model to consider groups with fewer parameters, leading to more efficient parameter utilization. The model ultimately learns to trade off between minimizing cross-entropy and reducing parameter-related costs. The group-wise loss is formulated as:

$$L_G = \alpha_G \sum_{i=1}^{N_g} \frac{W_i}{W_{max}} f_i^G p_i^G \tag{12}$$

$$f_i^{Grp} = \frac{N_g}{K_g} \sum_{t=1}^T \mathbb{1}(GS_{i,t} \in topk(GS_t)) \tag{13}$$

$$p_i^G = \frac{1}{T} \sum_i^T s_{i,t}^{G\ '} \tag{14}$$

$$s_{i,t}^{G\ '} = \frac{GS_{i,t}}{\sum_j^{N_g} GS_{j,t}} \tag{15}$$

where $W_i$ is the parameter count of group $i$, $f^G i$ is the group's routing frequency, the balance factor $\alpha_G$ is assigned an extremely small value and $p_i^G$ is its average normalized routing score.

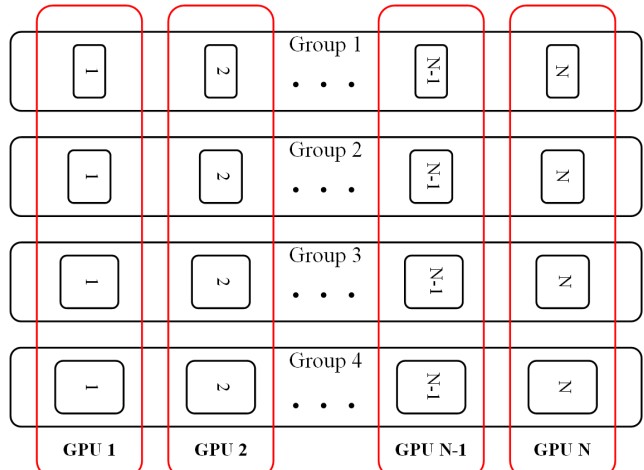

Figure 2: An example of All-size Group-decoupling Allocation.

## 3.5 LOAD BALANCE CONSIDERATION

Experts with larger hidden dimensions (i.e., those exceeding a base width $W_{\text{base}}$) introduce disproportionately higher memory and computational costs. If not carefully managed, this imbalance can lead to severe GPU load imbalances, where certain GPUs become bottlenecks while others remain underutilized. This inefficiency hampers overall training performance and scalability. To mitigate this issue, we introduce **All-size Group-decoupling Allocation** and **Intra-Group Experts Auxiliary Loss**, which work synergistically to achieve a uniform distribution of computational load across GPUs, thus ensuring balanced resource utilization.

### 3.5.1 ALLOCATION STRATEGY

An All-size expert set consists of the $i$-th expert from all groups. Each GPU is assigned multiple such sets, ensuring that the total number of expert parameters on each GPU remains consistent and smoothing out the variance in parameter size across the system. If expert workloads are evenly balanced within each group (which is encouraged by our auxiliary loss design), this approach leads to balanced GPU utilization overall.

As illustrated in Fig. 2 (with $N_g = 4$), each GPU hosts one All-size expert set (e.g., experts $E_{1,i}, E_{2,i}, E_{3,i}, E_{4,i}$). Regardless of the group selection during routing, as long as expert activation within each group is balanced, overall GPU resource usage remains evenly distributed.

### 3.5.2 INTRA-GROUP EXPERTS AUXILIARY LOSS

In addition to the standard cross-entropy loss, we incorporate an intra-group experts auxiliary loss $L_E$ adapted from DeepSeekV2 (Liu et al., 2024a) to encourage balanced expert usage during routing. While DeepSeekV2 penalizes imbalance across all experts globally, our approach focuses on experts within each selected group, promoting uniform routing frequencies locally. This design ensures that all experts within an active group are selected with equal frequency during training, leading to better load distribution across GPUs.

The auxiliary loss is defined as:

$$L_E = \alpha_E \sum_{g=1}^{N_g} \sum_{i=1}^{N} f_{g,i}^E p_{g,i}^E \tag{16}$$

$$f_{g,i}^E = \frac{N}{K_e} \sum_{t=1}^{T} \mathbb{1}(ES'_{g,i,t} \in topK_e(ES'_t)) \tag{17}$$

$$p_{g,i}^E = \frac{1}{T} \sum_{i}^{T} S_{g,j,t}^{Exp} \tag{18}$$

$$S_{g,j,t}^E = \frac{ES'_{g,i,t}}{\sum_{j}^{N} ES'_{g,j,t} + \epsilon} \tag{19}$$

where $f_{g,i}^E$ represents the normalized routing frequency of the $i$-th expert in group $g$, $s_{g,i,t}^{e}{}'$ is the normalized routing score, $p_{g,i}^E$ is the average selection probability across time steps, the balance factor $\alpha_E$ is assigned an extremely small value and $\epsilon$ is a very small constant to ensure that the denominator is not 0.

## 4 EXPERIMENTS

### 4.1 EXPERIMENTAL SETUP

**Compute Infrastructure.** All models were trained on a 16-node GPU cluster, with each node equipped with eight NVIDIA GPUs. We used the Megatron-LM framework (Shoeybi et al., 2019) to implement our MoHGE variants, as well as the dense and MoE baseline models.

**Pretraining Data.** Our pretraining corpus was created by merging and deduplicating three large English datasets: DataComp-LM, FineWeb, and The Pile. The combined corpus underwent standard noise filtering and quality checks to ensure data integrity. For all experiments, we sampled 0.58 trillion tokens from this cleaned, unified corpus.

**Model Configurations.** We evaluated three Transformer variants at the 1B 3B, and 14B parameter scales: a Dense model whose parameters are equal to the active parameters of the MoE baseline, a uniform-expert MoE baseline, and our proposed MoHGE architecture with heterogeneous expert groups. The MoE baseline is adapted from DeepSeekV2 (Liu et al., 2024b), with hyperparameters adjusted to align parameter counts across models for fair comparison. Detailed architectural configurations for all evaluated models are summarized in **APPENDIX**.

**Training Hyperparameters.** Each MoE model was trained for 2 full epochs on the 0.58 trillion–token corpus, using a fixed sequence length of 4,096. We used the AdamW optimizer with $\beta_1 = 0.9$, $\beta_2 = 0.95$, and a weight decay of 0.1. A cosine-decay learning rate schedule was applied, starting at $3 \times 10^{-4}$ and annealing to a minimum of $3 \times 10^{-5}$.

### 4.2 MAIN RESULTS

Following OpenCompass protocols (Contributors, 2023), Table 1 reports the zero-shot or few-shot (Kojima et al., 2022; Brown et al., 2020) in-context learning performance of our pretrained MoHGE models on a diverse suite of downstream tasks, including MMLU (Hendrycks et al., 2020), SIQA (Sap et al., 2019), GSM8K (Cobbe et al., 2021), LAMBADA (Paperno et al., 2016), MATH (Hendrycks et al., 2024), PIQA (Bisk et al., 2020) Bisk et al. (2020) and TriviaQA (Joshi et al., 2017).

As reported in Table 1, averaged over three evaluate runs, MoHGE consistently outperforms both conventional MoE and dense models across all scales, achieving state-of-the-art results on several benchmarks. Compared to the MoE baseline, MoHGE achieves a more favorable trade-off between parameter efficiency and downstream performance by activating fewer expert parameters while simultaneously requiring fewer total parameters.

Specifically, MoHGE reduces the overall parameter count by nearly 20% relative to standard MoE, and the number of activated parameters in the expert layer is reduced by approximately one quarter. This substantial reduction highlights its effectiveness in balancing model capacity with efficiency.

| Method | Total Parameters | Activated Parameters of Experts | MMLU | SIQA | GSM8K | LAMBADA | MATH | PIQA | TriviaQA |
|--------|-----------------|-------------------------------|------|------|-------|---------|------|------|----------|
| Dense | 0.570B | – | 25.41 | 34.93 | 1.79 | 51.87 | 1.22 | 44.85 | 25.05 |
| MoE-1B | 1.098B | 0.163B | 25.38 | 35.12 | 1.74 | 53.20 | 1.26 | 46.09 | **25.86** |
| MoHGE-1B | 0.891B | 0.122B | **25.98** | **35.17** | **1.97** | **53.75** | **1.29** | **48.85** | 25.71 |
| Dense | 0.807B | – | 26.36 | 35.30 | 2.79 | 61.02 | 1.33 | 47.35 | 34.98 |
| MoE-3B | 3.3614B | 0.376B | 26.22 | 35.41 | 3.03 | 60.86 | 1.34 | **49.08** | 39.16 |
| MoHGE-3B | 2.821B | 0.295B | **26.41** | **35.56** | **4.02** | **62.37** | **1.36** | **49.08** | **39.20** |
| Dense | 1.672B | – | 30.78 | 42.29 | 4.61 | 68.05 | 6.81 | 54.92 | 50.26 |
| MoE-14B | 16.760B | 1.191B | 31.18 | 44.28 | 4.88 | 67.94 | 7.29 | 56.71 | 51.77 |
| MoHGE-14B | 14.122B | 0.843B | **31.62** | **45.62** | **5.73** | **69.89** | **9.42** | **58.73** | **52.69** |

Table 1: Comparison between Dense model, MoE baseline and our MoHGE, the highest scores for each benchmark is highlighted in bold. MoHGE achieves slightly better performance while activating the fewest parameters. Furthermore, our model requires fewer total parameters than the baseline in addition to its efficiency advantages.

| Benchmark | MoE-1B(hours) | MoHGE-1B(hours) | MoE-3B(hours) | MoHGE-3B(hours) | MoE-14B(hours) | MoHGE-14B(hours) |
|-----------|---------------|-----------------|---------------|-----------------|----------------|------------------|
| MMLU | 6.90 | 6.77 | 9.85 | 9.58 | 19.27 | 18.86 |
| SIQA | 0.93 | 0.90 | 1.29 | 1.17 | 2.51 | 2.33 |
| GSM8K | 0.59 | 0.62 | 0.84 | 0.86 | 1.63 | 1.62 |
| LAMBADA | 2.24 | 2.22 | 3.17 | 3.03 | 6.27 | 6.08 |
| MATH | 2.24 | 2.23 | 3.18 | 3.02 | 6.33 | 6.09 |
| PIQA | 0.85 | 0.78 | 1.20 | 1.09 | 2.38 | 2.17 |
| TriviaQA | 4.11 | 3.95 | 5.78 | 5.46 | 11.46 | 10.85 |

Table 2: The inference duration of the MoE and MoHGE models on downstream tasks.

The inference times are demonstrated in Table 2. Regarding the slight increase in inference time on GSM8K which is a complex mathematical reasoning task, our routing analysis reveals that the MoHGE tends to select expert groups with larger parameter on GSM8K and this achieves higher accuracy while resulting in more inference time. Altogether, our model achieves relatively faster inference speeds, showing superior inference efficiency.

## 4.3 ABLATION STUDY ON AUXILIARY LOSS COEFFICIENTS

We conduct an ablation study to analyze the effect of different auxiliary loss coefficients on model performance. A coefficient of 0 indicates the absence of the auxiliary loss.

As shown in Table 3, the intra-group experts auxiliary loss yields a modest performance gain and setting $\alpha_{Exp} = 2.5e - 3$ achieves better results. Combining it with the group-wise auxiliary loss further improves results. Although the group-wise loss contributes only marginally to accuracy, it reduces the number of activated parameters. Based on the trade-off between evaluation performance and computational efficiency, we find that setting $\alpha_{Exp} = 2.5e - 3$ and $\alpha_{Grp} = 1e - 4$ enables the our models to achieve an optimal balance.

## 4.4 ANALYSIS ON TOKEN ROUTING

### 4.4.1 ROUTING ANALYSIS OF LOSS FUNCTION

Building on the optimal configurations identified in Table 3, we conduct experiments on two model configurations:

**Utilizing only intra-group expert auxiliary loss**: $\alpha_{Exp} = 2.5e - 3$ and $\alpha_{Grp} = 0$.

**Combining two loss functions**: $\alpha_{Exp} = 2.5e - 3$ and $\alpha_{Grp} = 1e - 4$.

We statistically analyzed the distribution of 100 million token routes across these configurations. As shown in Fig. 3, the overall route distribution does not exhibit concentration in specific groups under either setup. However, introducing the group routing loss shifts the token routing behavior: instead of predominantly favoring larger expert groups, tokens are distributed toward smaller. This indicates that the group-wise loss encourages the selection of smaller expert groups which can accommodate the current task difficulty in condition of relatively uniform routing distribution.

| Model | $\alpha_{Exp}$ | $\alpha_{Grp}$ | Activated Parameters of Experts | MMLU | SIQA | PIQA | LAMBADA | TriviaQA |
|---|---|---|---|---|---|---|---|---|
| MoHGE-1B | 0 | 0 | 139M | 25.43 | 34.73 | 47.62 | 52.20 | 25.03 |
| | 2.5e-3 | 0 | 132M | 25.61 | 34.82 | 47.93 | 53.35 | 25.37 |
| | 5e-3 | 0 | 131M | 25.87 | 34.74 | 48.77 | 53.14 | 25.20 |
| | 2.5e-3 | 1e-4 | 122M | **25.98** | **35.17** | 48.85 | **53.75** | **25.42** |
| | 2.5e-3 | 1e-3 | 122M | 25.94 | 35.10 | 48.28 | 52.99 | 25.25 |
| | 5e-3 | 1e-4 | 119M | 25.96 | 34.86 | 48.12 | 53.16 | 25.39 |
| MoHGE-3B | 0 | 0 | 324M | 25.88 | 35.29 | 48.65 | 61.37 | 38.01 |
| | 2.5e-3 | 0 | 307M | 26.11 | 35.45 | 48.53 | 61.62 | 38.53 |
| | 5e-3 | 0 | 310M | 26.03 | 35.32 | 48.67 | 61.75 | 38.45 |
| | 2.5e-3 | 1e-4 | 295M | 26.31 | **35.56** | **49.08** | **62.37** | **39.20** |
| | 2.5e-3 | 1e-3 | 297M | **26.36** | 35.12 | 48.83 | 62.10 | 38.68 |
| | 5e-3 | 1e-4 | 289M | 26.27 | 35.47 | 48.21 | 61.85 | 38.51 |
| MoHGE-14B | 0 | 0 | 897M | 31.37 | 44.92 | 57.94 | 68.57 | 51.72 |
| | 2.5e-3 | 0 | 884M | 31.18 | 45.03 | 58.07 | 68.95 | 51.86 |
| | 5e-3 | 0 | 875M | **31.71** | 45.39 | 58.27 | 68.90 | 52.29 |
| | 2.5e-3 | 1e-4 | 843M | 31.62 | **45.62** | **58.73** | **69.89** | 52.49 |
| | 2.5e-3 | 1e-3 | 854M | 30.87 | 45.07 | 58.22 | 69.10 | **52.75** |
| | 5e-3 | 1e-4 | 859M | 31.38 | 44.78 | 58.15 | 69.85 | 51.67 |

Table 3: The evaluation results for varying coefficients of the auxiliary loss function. The highest-performing score for each benchmark is highlighted in bold, while the second-highest score is underlined.

| | GPU_1 | GPU_2 | GPU_3 | GPU_4 | GPU_5 | GPU_6 | GPU_7 | GPU_8 | Avg | Std |
|---|---|---|---|---|---|---|---|---|---|---|
| Group_1 | 2.05M | 1.89M | 1.92M | 1.99M | 2.02M | 1.86M | 1.97M | 2.01M | 1.96M | 0.06283 |
| Group_2 | 1.66M | 1.74M | 1.82M | 1.62M | 1.59M | 1.74M | 1.58M | 1.75M | 1.69M | 0.08166 |
| Group_3 | 1.58M | 1.59M | 1.67M | 1.50M | 1.51M | 1.52M | 1.64M | 1.70M | 1.59M | 0.07115 |
| Group_4 | 1.67M | 1.58M | 1.65M | 1.59M | 1.67M | 1.62M | 1.70M | 1.51M | 1.62M | 0.05786 |
| Group_5 | 1.25M | 1.39M | 1.38M | 1.27M | 1.33M | 1.38M | 1.35M | 1.26M | 1.33M | 0.05452 |
| Group_6 | 1.45M | 1.41M | 1.45M | 1.31M | 1.49M | 1.40M | 1.50M | 1.41M | 1.43M | 0.05629 |
| Group_7 | 1.52M | 1.60M | 1.57M | 1.43M | 1.41M | 1.55M | 1.62M | 1.43M | 1.52M | 0.07745 |
| Group_8 | 1.39M | 1.32M | 1.46M | 1.41M | 1.33M | 1.35M | 1.32M | 1.36M | 1.37M | 0.04630 |

Table 4: For 14B scale model, the number of tokens routed to each GPU roughly closes to the average value.

### 4.4.2 ROUTING ANALYSIS OF GPU UTILIZATION

To rigorously evaluate the balancing of GPU utilization, we conduct a GPU-level assessment for 14B scale model by strategically assigning the $i$-th expert from each capacity group to the $i$-th GPU. This experimental design allows us to precisely track how tokens are distributed across experts of varying sizes on each GPU, which reflects the frequency of token processed by experts of different sizes on each GPU. Table 4 shows that experts of uniform size receive nearly equal routing frequencies across GPUs, indicating balanced intra-group expert and GPU utilization. This confirms that our All-size Group-decoupling Allocation and Intra-group Experts Auxiliary Loss effectively maintain equilibrium in both computational resource loading and expert activation patterns.

## 5 RELATED WORK

The MoE model was originally proposed by Jacobs et al. (1991). Subsequently, Shazeer et al. (2017) introduced Sparsely-Gated Mixture-of-Experts which demonstrate substantial improvements in model capacity and efficiency. Furtherly, SwitchTransformer, proposed by Fedus et al. (2022), incorporated MoE into the Transformer architecture's Feed-Forward Network layers with simplified MoE routing algorithm, showing great potential in large-scale Transformer models. Typically, MoE models consist of homogeneous experts, each with identical number of parameters, and a predetermined number of experts are activated regardless of the input's complexity. However, this hinders effective expert specialization and efficient parameter utilization.

Huang et al. (2024) proposed Top-P routing algorithm to address inefficient parameter utilization by assigning different numbers of experts to different tokens. Nevertheless, this method relies on fixed threshold settings and employs a rudimentary approach to difficulty modeling, making it challenging to adapt effectively to diverse inputs. Sun et al. (2024) proposed the Diverse Size Experts structure for each FFN layer, where each expert has a different parameter size to handle generating tasks of varying difficulty. However, they employ a uniform routing strategy that fails to route input tokens to the most suitable expert, resulting in inefficient parameter utilization and compromised perfor-

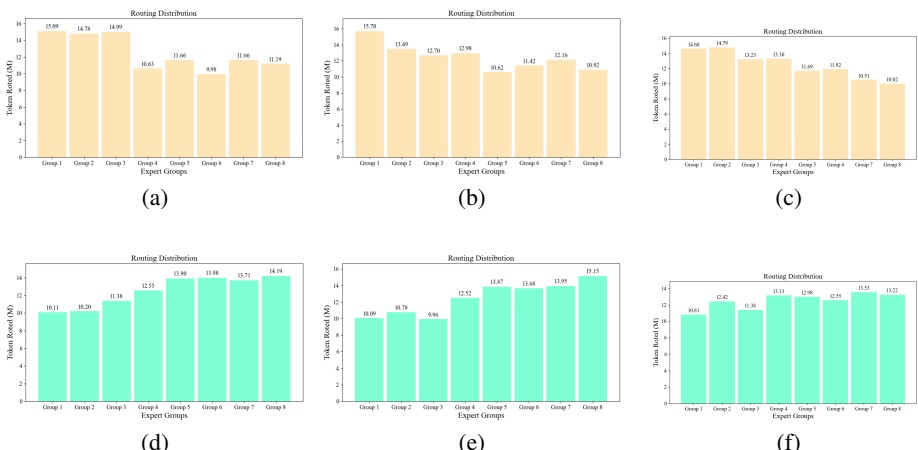

Figure 3: The number of tokens routed to each expert group. (a) Our MoHGE-1B with Group-Wise Auxiliary Loss. (b) Our MoHGE-3B with Group-Wise Auxiliary Loss. (c) Our MoHGE-14B with Group-Wise Auxiliary Loss. (d) Our MoHGE-1B without Group-Wise Auxiliary Loss. (e) Our MoHGE-3B without Group-Wise Auxiliary Loss. (f) Our MoHGE-14B without Group-Wise Auxiliary Loss.

mance. To enhance effective expert specialization and efficient parameter utilization, Wang et al. (2024) proposed the Heterogeneous Mixture of Experts with the parameter penalty loss to encourage the activation of smaller experts. They furtherly explored three types of heterogeneity structures and showed the hybrid structure that jointly combines both homogeneous and heterogeneous offers better performance than completely heterogeneous structure, such as Geometric and Arithmetic sequence structures. However, this kind of hybrid structure suffers from a problem of unbalanced computation and communication arising from the heterogeneous nature of experts, which lead to inefficient training and restrict its scalability, more optimal setups for hybrid structure need to be explored.

In contrast, our work introduces a general hybrid heterogeneous MoE architecture that partitions experts into heterogeneous groups, where each group consists of homogeneous experts. We propose the two-level routing mechanism and Group-Wise Auxiliary Loss to enable the gating model to select expert groups with varying parameter sizes based on token difficulty, thus improving parameter utilization. Additionally, we propose the All-size Group-decoupling Allocation strategy, which ensures a uniform distribution of parameters across GPUs, facilitated by the Intra-Group Experts Auxiliary Loss, ensuring balanced GPU utilization.

# 6    CONCLUSION

In this work, we propose MoHGE, a novel Mixture-of-Experts (MoE) architecture that introduces group-wise expert size variation to better accommodate the diverse complexity of token predictions. We introduce a two-level routing strategy, coupled with a Group-Wise Auxiliary Loss, to enable the selection of expert groups with suitable parameter sizes based on task difficulty. To address GPU load imbalance, we propose the All-size Group-decoupling Allocation strategy, which allocates an equal number of experts from each group to each GPU, ensuring balanced GPU memory usage. And we apply the Intra-Group Experts Auxiliary Loss to maintain balanced routing probabilities within each group, promoting uniform expert activation and GPU utilization. Our experimental results demonstrate that MoHGE achieves improved performance while reducing both the total number of parameters and the number of activated parameters. Detailed routing analysis further confirms effective GPU utilization balance. By rethinking how expert capacities should vary and be allocated, MoHGE paves the way for developing more efficient and capable large language models.

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

# 7 APPENDIX

## 7.1 MODEL CONFIG

Detailed architectural configurations for all evaluated models are summarized in Table 5.

| Configuration | 1B Scale | 3B Scale | 14B Scale |
|---|---|---|---|
| **Shared Configuration** | | | |
| Transformer Layers | 9 | 15 | 36 |
| Input Dim | 1024 | 1024 | 1024 |
| Attention Heads | 16 | 16 | 16 |
| **Dense Model** | | | |
| FFN Hidden Dim | 4096 | 6144 | 8192 |
| **MoE Baseline** | | | |
| $N_e$ | 32 | 64 | 128 |
| $K_e$ | 6 | 6 | 6 |
| Shared Experts $N_s$ | 2 | 2 | 2 |
| Expert Hidden Dim | 832 | 1024 | 1280 |
| **MoHGE** | | | |
| $N_g$ | 8 | 8 | 8 |
| $K_g$ | 3 | 3 | 3 |
| $N_e$ | 32 | 64 | 128 |
| $K_e$ | 6 | 6 | 6 |
| Shared Experts $N_s$ | 2 | 2 | 2 |
| Hidden Dims of Expert Groups | {256, 320, 384, 512, 640, 768, 832, 896} | {384, 512, 640, 768, 896, 1024, 1152, 1280} | {640, 768, 896, 1024, 1152, 1280, 1408, 1536} |

Table 5: Architecture configurations of the evaluated models at both 1B, 3B and 14B parameter scales. MoHGE uses heterogeneous expert groups with different hidden dimensions.

## 7.2 ROUTING ANALYSIS OF TOKENS OF DIFFERENT DIFFICULTIES

| Token Ranks | Group 1 | Group 2 | Group 3 | Group 4 | Group 5 | Group 6 | Group 7 | Group 8 |
|---|---|---|---|---|---|---|---|---|
| Top 1K | 16.3% | 14.9% | 14.1% | 12.2% | 12.3% | 10.5% | 10.0% | 9.7% |
| Top 1K-5K | 15.0% | 14.4% | 13.4% | 13.1% | 12.4% | 10.3% | 10.9% | 10.5% |
| Top 5K-10K | 13.6% | 13.4% | 13.7% | 12.6% | 11.5% | 12.4% | 11.5% | 11.3% |
| Beyond 10K | 11.3% | 12.0% | 11.4% | 12.7% | 13.0% | 12.7% | 13.6% | 13.3% |

We categorized the vocabulary into four difficulty levels based on occurrence frequency ranks in training corpus: Top 1K (easiest), Top 1K-5K, Top 5K-10K and Beyond 10K (most difficult). Section 7.2 shows the ratios of tokens with different difficulty routed to different expert groups. These results demonstrate that simpler tokens tend to be routed to expert groups with fewer parameters, and this validates the effectiveness of our method.

