# OpenReview forum: "Mixture of Heterogeneous Grouped Experts for Language Modeling"
_ICLR.cc/2026/Conference — Submitted to ICLR 2026_

### Official Review · Reviewer_6p47 · 2025-10-29

**Soundness:** 3
**Presentation:** 2
**Contribution:** 2
**Rating:** 4
**Confidence:** 4

**Summary:**

Traditional Mixture-of-Experts LLM design assumes all experts to be of the same amount of parameters. Recent findings show that this could lead to a waste of computation, as different tokens have different difficulties, hence require different amounts of computation to process. Existing explorations in this direction either assume all experts should be treated equally during routing, or suffer from GPU utilization imbalance. The author proposes MoHGE, which introduces a two-layer routing schema, to allow better GPU utilization balance and more diverse routing. Empirical results show that the proposed method outperforms the baseline dense model and training model.

**Strengths:**

- The design is intuitive and well-motivated, making the core idea easy to grasp.

- The method section (sec 3) is clearly written, which is helpful for the reader to understand the approach.

- The paper provides appropriate background material, achieving a good balance between necessary context and conciseness without digressing into unrelated details.

**Weaknesses:**

- Insufficient experiment and baseline selection. Only the dense model and traditional MoE model was selected for comparison. In my opinion, the author should **compare against the two baselines** mentioned on line 47, which are **MoDSE and HMoE**. However, there is not a direct comparison against these two baselines in the experimental section. The main result only shows that the proposed method is better than the traditional MoE or a dense model, yet the advantage of heterogeneous experts has already been described in aforementioned prior works. Without a clear comparison against these baselines, it's hard to convince the reader that it is the proposed change that improves the performance. Specifically:
  - What if we utilize a global load balancing loss, like in MoDSE? Is there a row in table 3 that is equivalent to such a setting? If so, why is it not clearly mentioned?
  - How bad would the load balance become if the author chooses to adopt the expert setting in HMoE? How would that affect the hardware utilization in your settings? What would be the difference in terms of the **batch size without OOM/GPU utilization/per iteration latency**?

- Insufficient details regarding the hardware and other training/inference setup. The author describes the GPU used as "NVIDIA GPUs", and other details are also unclear, such as the batch size chosen for training or inference. Is any inference specific acceleration technique being used? What is the communication kernel used for MoE routing? **Code has not been provided.**

- Minor writing issues:
  - Font is way too small at many places.  This creates difficulty for the readers and please consider improving the readability. I found it very difficult to read after printing it out and I had to use a large screen to read this paper.
    - All equations
    - Figure 1 & 3 Caption
  - Line 156: incorrect linewidth.
  - Missing "," on line 297.

**Questions:**

See weaknesses for a plethora of questions. Besides:
- Why were two epochs being trained on the LLM? Doesn't the second epoch lead to overfitting?

- What is the inference/training time for the dense model? Given that the dense models are typically faster than MoE when the amount of parameters are equal, I wonder whether there exists a significant advantage to choose MoE model under the described scenario.

- One interesting design choice is that for each token, the final activated expert is selected top-$K_e$ experts globally, instead of always selecting a certain number of experts from each group, which theoretically has an even better hardware balance. Is there any empirical evidence for the design?

---

> ### Author Response · Authors · 2025-11-19
>
> W1:
>
> We thank the reviewer for pointing this out. We acknowledge the importance of comparing against MoDSE and HMoE. Unfortunately, neither of these methods has publicly released their code or model implementations, which makes a direct and fair empirical comparison infeasible at this stage.
>
> - There is no row in Table 3 that is equivalent to this setting. We will supplement the global balancing method.
>
> -  Since the HMoEd code is not released, we are unable to resolve your this question.
>
> W2:
>
> We will add details about the hardware and other training/inference settings.
> All models were trained on a 16-node GPU cluster, with each node equipped with eight NVIDIA H800 Tensor Core GPU.
> The batch size for training was set to 256
> We used FlashAttention to accelerate model training and inference.
>
> W3:
>
> We thank the reviewer for highlighting these writing and formatting issues. We will carefully correct the identified errors, including equation formatting, figure captions, and punctuation. In addition, we will appropriately increase the font size throughout the paper to ensure better readability in both digital and printed formats.
>
> Q1:
>
> After training on our dataset for one epoch, we found that the results did not meet our expectations. After adding another epoch, we found that the model performance improved, so we standardized it to two epochs.
>
> Q2:
>
> We thank the reviewer for the question. We have included the inference time of the dense models in the followed table. We would like to clarify that when comparing models with a similar number of activated parameters (rather than total parameters), the dense models indeed run slightly faster than MoE models, as expected. However, the performance gap remains small and within an acceptable range. Considering that MoE achieves superior performance with fewer activated parameters, we believe the trade-off still favors the MoE design in this setting.
>
> |    | 1B | 3B | 14B |
> | ----- | ----- | ----- | ----- |
> | MMLU | 	6.22	| 9.06 | 17.83 |
> | SIQA | 0.85 | 1.09	| 2.17 |
> | GSM8K | 0.54 | 0.77 | 1.52 |
> | LAMBADA | 2.10 | 2.83 | 5.70 |
> | MATH | 2.12 | 2.86 | 5.74 |
> | PIQA | 0.79 | 1.03 | 2.07 |
> | TriviaQA | 3.87 | 5.24 | 10.39 |
>
>
>
> Q3:
>
> We thank the reviewer for the insightful question. We conducted empirical studies at the 1B scale to compare the two routing strategies: (1) globally selecting the top-6 experts across all 3 expert groups, and (2) first selecting 3 expert groups and then choosing the top-2 experts within each group. The resulting validation losses were 2.25 and 2.28, respectively. Given the consistently better performance of the global top-K routing, we adopted configuration (1) in our final design.

---

### Official Review · Reviewer_3vJN · 2025-10-30

**Soundness:** 2
**Presentation:** 2
**Contribution:** 3
**Rating:** 4
**Confidence:** 4

**Summary:**

To enable models to dynamically match computing resources based on token-specific needs, the development of Heterogeneous MoE is of great significance. Existing studies on Heterogeneous MoE either suffer from limited performance and low parameter utilization efficiency, or face the issue of unbalanced GPU utilization, with a universal heterogeneous MoE architecture still lacking. This paper proposes the Mixture of Heterogeneous Grouped Experts (MoHGE) — an innovative MoE architecture that introduces a two-level routing mechanism, enabling more refined and efficient expert selection based on the characteristics of each input token. This paper also proposes the "Group-Wise Auxiliary Loss", which improves parameter utilization efficiency without compromising model performance, including the All-size Group-decoupling Allocation strategy and Intra-Group Experts Auxiliary Loss.

**Strengths:**

This paper proposes the Mixture of Heterogeneous Grouped Experts (MoHGE) — an innovative MoE architecture that introduces a two-level routing mechanism, enabling more refined and efficient expert selection based on the characteristics of each input token. This paper also proposes the "Group-Wise Auxiliary Loss", which improves parameter utilization efficiency without compromising model performance, including the All-size Group-decoupling Allocation strategy and Intra-Group Experts Auxiliary Loss.

**Weaknesses:**

The predefined grouping of experts in the authors' proposed Mixture of Heterogeneous Grouped Experts (MoHGE) leads to a significant reduction in the combinatorial diversity of experts during routing.

All comparative experiments conducted by the authors lack comparisons with existing research on Heterogeneous MoE (including HMoE), only comparing against MoE models and Dense models—among these, the comparison with Dense models is unnecessary. Furthermore, in experimental design, the authors should strive to ensure consistency in total parameters and activated parameters; currently, these two critical factors exhibit significant discrepancies, making the experiments insufficiently rigorous.

When scaling the model, will the grouping mechanism of MoHGE affect the expert parallelism (ep) strategy?

There is a lack of rigorous ablation experiments on MoHGE.

It is anticipated that refining the experimental section would result in a higher rating.

**Questions:**

Refer to Weaknesses

---

> ### Author Response · Authors · 2025-11-19
> **To Reviewer 3vJN**
>
> W1：
>
> We appreciate the reviewer’s concern. While predefined grouping indeed reduces the theoretical combinatorial space of expert combinations, most of these combinations are not meaningful or beneficial for the target task. Our design applies two routing stages, which effectively identify and activate the most suitable expert groups for the given input. In practice, this hierarchical routing preserves the useful diversity while avoiding inefficient or suboptimal expert combinations, leading to more stable and task-relevant expert selection.
>
> W2：
>
> We thank the reviewer for the valuable feedback. We apologize for not including the heterogeneous MoE baseline HMoE, as its code has not been released.Although the total and activated parameters are not exactly matched across all baselines, they remain within the same order of magnitude. Under this setting, our method consistently achieves superior performance while using fewer total parameters and fewer activated parameters.
>
> W3:
>
> We thank the reviewer for the question. The grouping mechanism in MoHGE does not interfere with the expert parallelism (ep) strategy. Since the selection of expert groups and the selection of experts within each group are decoupled, scaling the model size does not introduce additional dependency across experts. Therefore, the expert parallelism strategy remains unaffected when scaling up the model.
>
> W4:
>
> We will further supplement the ablation experiments.

---

### Official Review · Reviewer_aBiB · 2025-10-31

**Soundness:** 2
**Presentation:** 2
**Contribution:** 2
**Rating:** 2
**Confidence:** 3

**Summary:**

The paper proposes Mixture of Heterogeneous Grouped Experts (MoHGE), a variant of the Mixture-of-Experts (MoE) framework that partitions experts into groups of varying sizes and introduces a two-level routing mechanism (group selection followed by intra-group expert routing). Two auxiliary objectives are added to encourage balanced expert usage: a Group-Wise Auxiliary Loss and an Intra-Group Experts Auxiliary Loss.
 Experiments on several language understanding and reasoning benchmarks (MMLU, GSM8K, SIQA, LAMBADA, PIQA, TriviaQA, MATH) across 1B, 3B, and 14B models show similar or slightly better accuracy compared to homogeneous MoE baselines, with roughly 20% fewer parameters.

**Strengths:**

- Addresses a practical bottleneck in large-scale MoE training.
- Method is straightforward and compatible with existing frameworks.
- Writing and figures are clear and professional.
- Experiments span multiple scales and include ablations on loss terms.

**Weaknesses:**

- Missing quantitative evidence of GPU utilization or efficiency.
- Reported improvements are small and lack statistical validation.
- No theoretical  analysis of routing dynamics or convergence.
- Experimental reporting omits critical hyperparameters and setup details.
- Contribution is incremental relative to prior heterogeneous MoE work.

---
1. Quantify “Efficiency” Claims (Page 3–4)
- Provide actual GPU load statistics, throughput (tokens/sec), and communication cost.
- Report training time per step and FLOPs vs. MoE baseline.
- Showing real efficiency data would validate the motivation and significantly raise the Quality and Significance scores.
2. Report Statistical Significance (Page 6–7, Table 1)
- Repeat experiments with multiple random seeds.
- Include standard deviation or confidence intervals.
- Apply paired t-tests to confirm that improvements are not noise.
3. Ablate Grouping vs. Routing (Page 5)
- Isolate the effect of heterogeneous expert sizing and two-level routing separately.
- Provide visualization of token-to-group assignment entropy.
- Helps clarify where gains actually come from, strengthening Originality.
4. Provide Theoretical or Analytical Justification (Page 4)
- Offer a short analysis on expected load variance under two-level routing.
- Discuss whether routing convergence differs from standard MoE gating.
5. Expand Experimental Reporting
- Include training setup: batch size, gradient accumulation, GPU count, communication strategy, optimizer, etc.
- Report training hours and memory footprint.
6. Broaden Evaluation Scope (Page 7–8)
- Add large-scale pretraining or autoregressive datasets (e.g., C4, Pile).
- Report cost per token to demonstrate real-world scalability.
7. Tone Down Overstatements (Page 1 & 9)
- Replace “establishes a new paradigm” with more measured phrasing like “offers a simple variant that modestly improves efficiency.”

**Questions:**

Although the goal—improving parameter efficiency and GPU load balance in MoE systems—is relevant, the evidence provided does not convincingly support that MoHGE achieves this goal in a meaningful way.
1. The motivation is not empirically substantiated.
 The paper claims that homogeneous experts lead to “severe GPU imbalance” and “inefficient utilization,” yet no quantitative evidence (e.g., utilization variance, expert activation frequency, or throughput) is reported. The claim remains speculative.
2. Efficiency claims lack measurements.
 The reported 20% reduction in parameters simply reflects reduced hidden dimensions for some expert groups (Page 6, Table 1). There is no analysis of actual computation cost, latency, or memory use. Without such data, it is unclear whether MoHGE is truly more efficient.
3. Empirical improvements are small and inconsistent.
 Across benchmarks, improvements over MoE are ≤1 point—likely within training variance (Table 1, Page 6). On GSM8K, inference is slower. Without multiple runs or variance reporting, these gains are not statistically credible.
4. Limited analysis of mechanism and behavior.
 The two-level routing and auxiliary losses are introduced heuristically, without analyzing how they affect routing stability or specialization. Figures 2–3 show routing distributions but offer no interpretation.
5. Incremental novelty.
 Similar ideas appear in HMoE and MoDSE. MoHGE reorganizes them hierarchically but does not introduce a fundamentally new concept or analytical perspective.

---

### Official Review · Reviewer_BbM7 · 2025-11-10

**Soundness:** 3
**Presentation:** 2
**Contribution:** 2
**Rating:** 4
**Confidence:** 3

**Summary:**

This paper introduces a new Mixture-of-Experts (MoE) architecture with different expert sizes. A two-level routing strategy is used to select experts based on task difficulty. Multiple techniques (group-wise auxiliary loss, all-size group-decoupling allocation strategy, and intra-group experts auxiliary loss) are utilized for addressing GPU load imbalance, routing imbalance issues. Extensive experiments are conducted to show the effectiveness of the proposed approaches.

**Strengths:**

1. The idea of the two-level routing strategy is intuitive and interesting.
2. Extensive experiments (1B, 3B, 14B models with 0.58T tokens) are conducted to show the effectiveness of the proposed methods.

**Weaknesses:**

1. Multiple loss functions are introduced in the pretraining stage, which makes the hyperparameter selection costly.
2. The design of group-wise auxiliary loss is not that clear to me. Why is routing to the larger experts a problem? The model performance improvement introduced by the group-wise auxiliary loss seems to be marginal. The authors claim that group-wise auxiliary loss reduces the number of activated parameters. Could the author provide more qualitative results?

**Questions:**

1. What is the intuitive reason for the design of equation 5?
2. What is the final loss function? Please list it explicitly.

---

> ### Author Response · Authors · 2025-11-19
> **To Reviewer BbM7**
>
> W1:
>
> We appreciate the reviewer’s comment. We acknowledge that incorporating two loss functions introduces an additional hyperparameter to tune, which slightly increases the complexity of hyperparameter selection. However, this overhead remains modest in practice, and the tuning cost associated with the two loss coefficients is manageable.
>
> W2:
>
> We thank the reviewer for the insightful question. Larger experts typically achieve stronger predictive performance, which leads the routing mechanism to preferentially select them. However, for relatively simple prediction cases, the performance gap between large and small experts is minimal, making this preference unnecessary and suboptimal in terms of parameter activation. Our group-wise auxiliary loss is designed to mitigate this bias by encouraging a more balanced expert utilization. As shown in Table 3, applying the group-wise auxiliary loss reduces the number of activated parameters by around 10% compared to the model without this loss, while maintaining comparable performance.
>
> Q1:
>
> We can determine the total number of parameters for all experts by setting an average expert hidden dimension $W_{base}$ with  the design of equation 5.
>
> Q2:
> The final loss function is
>
> $L_{final} = L_{task} + L_G + L_E$
>
> Where $L_{task}$ represents the loss of the main task, such as the prediction loss of the next token.

---

### Meta-Review · Area_Chair_cEsJ · 2026-01-02

**Summary:**

### Strength emphasized by reviewers
1. The idea of the two-level routing strategy is intuitive.
2. The paper is clearly written, easy to understand.
3. The paper provides appropriate background material.

### Ensemble of concerns and suggestions, including my comments
1. **Experiments**:
    1. Lacking comparison with existing heterogenous work, e.g., MoDSE, HMoE. [Reviewer 3vJN, 6p47]. This is necessary for this paper, as the paper’s claims existing heterogenous work suffers from insufficient parameter utilization and training efficiency as the motivation.
    2. Ablation is not enough [Reviewer 3vJN, 6p47]. For example, (1) group-wise + intra-group expert loss V.S. global load balancing loss, (2) the way of selecting the final activated experts, global scoring and selection V.S. balanced group selection, and so on.
    3. Multiple runs w. different random seeds are needed for demonstrating the advantage.
2. **Practicality (easy applicable or not)** [Reviewer BbM7]: Heterogeneous expert design introduces additional complexity, including additional hyper-parameter tuning.
3. **Practicality (performance-efficiency curve)**: (1) As dense model inference is faster when #param is equal, is the heterogeneous design really an advantage [Reviewer 6p47]. (2) The comparison should ensure consistency in total parameters and activated parameters [Reviewer 3vJN]. *For these two questions, I recommend the authors to provide  performance-efficiency curves (w.r.t. inference time, #param, etc.).* Besides, (3) the work designs a group-wise auxiliary loss to slightly penalize expert groups with larger parameter sizes, did this parameter efficiency really correspond to actual inference efficiency?
4. **Justification of design choices should be made more clear** [Reviewer 3vJN]: For example, predefined grouping of experts leads to a significant reduction in the combinatorial diversity of experts during routing, need further justification (e.g., theoretical analysis or experiment) on why this reduction won’t limit the performance. Also see "1. Experiments -2" for other design choices that need justification.
5. **Other**: Insufficient detail on hardware and other training/inference setup; Need to provide the dense model’s inference time as reference; Some formatting issue. [Reviewer 6p47]

**Reviewer Concerns:**

**Addressed**: For "5. Other" proposed by Reviewer 6p47, in the rebuttal, the authors provided the hardware and training/inference setup, as well as the dense model's inference time. The authors didn't submit the revision, so the formatting issue is not fixed yet (but should be easy).

**Unaddressed, Important**: For "1. Experiment - 1, lacking comparison with existing heterogenous work", I think this is necessary for this work. The authors claim that due to the prior work not open source their code, they cannot provide the comparison. I think with careful implementation, the authors could and should implement these essential baselines in a comparable way using their own setting. Even the prior work doesn't open source their original work, this efforts will provide a valid baseline comparison and also a solid ablation as well.

**Still partially unaddressed**: The rebuttal does not provide additional ablation experiments. The rebuttal explains why the work uses global scoring and selection instead of balanced group selection for Reviewer 6p47, and the reduction of the combinatorial diversity won't degrades the performance for Reviewer 3vJN, and so on. Nevertheless, the explanations are not convincing enough in my view.

**Reviewer Scores:**

I think Reviewer BbM7 might increase the score to 6 at most, other reviewers won't increase their scores.

---

### Decision · Program_Chairs · 2026-01-26

Reject